# Genetic Diversity and Population Structure of an Arctic Tertiary Relict Tree Endemic to China (*Sassafras tzumu*) Revealed by Novel Nuclear Microsatellite (nSSR) Markers

**DOI:** 10.3390/plants11202706

**Published:** 2022-10-13

**Authors:** Shuang Wang, Ying Wang, Jingbo Zhou, Pan Li, Hungwei Lin, Ye Peng, Lipeng Yu, Yunyan Zhang, Zhongsheng Wang

**Affiliations:** 1College of Life Sciences, Nanjing University, Nanjing 210023, China; 2Systematic & Evolutionary Botany and Biodiversity Group, MOE Laboratory of Biosystem Homeostasis and Protection, College of Life Sciences, Zhejiang University, Hangzhou 310058, China; 3College of Biology and the Environment, Nanjing Forestry University, Nanjing 210037, China; 4Mount Longwang Nature Reserve, Huzhou 313300, China

**Keywords:** *Sassafras tzumu*, Arctic Tertiary relict plant, nuclear microsatellite, genetic diversity, population structure, population differentiation, ESU

## Abstract

*Sassafras tzumu* (Hemsl.) Hemsl., as an Arctic Tertiary relict woody species, is an ecologically and economically important deciduous tree endemic to southern China. Nonetheless, the genetic resources and backgrounds of *S. tzumu* are still lacking and remain largely unclear. Here, we predicted 16,215 candidate polymorphic nuclear microsatellite (nSSR) loci from the assembled nucleus databases of six geographic-distant individuals of *S. tzumu* via CandiSSR. Among these nSSRs, the di- (75.53%) and tri-nucleotide (19.75%) repeats were the most abundant, and 27 new polymorphic SSRs were developed and characterized in 136 individuals from six natural populations of *S. tzumu*. The majority of the above 27 SSRs (24 loci, 88.89%) presented moderate polymorphism (mean *PIC* = 0.356), and the transferability of these markers in other *Sassafras* species was high (85.19%). A moderately low level of genetic diversity and a high variation (*F*_ST_ = 0.286) of six wild populations of *S. tzumu* were illuminated by 16 selected polymorphic nSSRs, with the average expected heterozygosity (*H*_E_) of 0.430 at the species level and *H*_E_ ranging from 0.195 to 0.387 at the population level. Meanwhile, a bottleneck effect was shown in two populations. Consistent with the results of the principal coordinate analysis (PCoA) and phylogenetic trees, structure analysis optimally divided these six *S. tzumu* populations into two clusters, and the further strong population subdivision appeared from *K* = 2 to *K* = 5, which corresponded to two evolutionarily significant units (ESUs). Moreover, the significant correlation between genetic and geographic distance was tested by the Mantel test (*r* = 0.742, *p* = 0.006), clarifying the effect about isolation by distance (IBD), which could be possibly explained by the low gene flow (*N*_m_ = 0.625), a relatively high degree of inbreeding (*F*_IS_ = 0.166), a relatively large distribution, and mountainous barriers. Above all, our research not only enlarged the useful genetic resources for future studies of population genetics, molecular breeding, and germplasm management of *S. tzumu* and its siblings but also contributed to proposing scientific conservation strategies and schemes for the better preservation of *S. tzumu* and other *Sassafras* (Lauraceae) species.

## 1. Introduction

*Sassafras* J. Presl (Lauraceae) contains three extant species with a disjunct distribution between East Asia and North America: *S. tzumu* (Hemsl.) Hemsl. is located in the Yangtze River basin and its southern areas in continental China; *S. randaiense* (Hayata) Rehder is merely distributed in Central Taiwan; whereas *S. albidum* (Nutt.) Nees is found in the east of North America [1,2]. Previous studies pointed out that *Sassafras* is monophyletic, and the species of North America (*S. albidum*) is sister to the clade of its two East Asian counterparts [3,4,5]. Morphologically, the above three sibling species mainly differ in their floral structure: *S. randaiense* distinguishes from *S. tzumu* in the number of anther chambers (two-locular versus four-locular), whereas we can tell *S. albidum* apart from its two Asian congeners in its six staminodes in the female flower and lacking the fourth staminal whorl [3,6].

*S. tzumu*, the focal species of our present study, is an Arctic Tertiary relict plant endemic to China, growing in both broad-leaved evergreen and mixed deciduous forests of subtropical China at an altitude range of 100–1900 m [7,8,9]. *S. tzumu* has strong tolerance for various types of soil, such as acid or sandy loam soil [7,8,10]. Additionally, *S. tzumu* has many important values in economy and ecology: its meticulous and durable timber is widely used for ship-building and manufacturing of fine furniture; its root and bark are used as medicine, with functions of activating blood circulation and dispersing blood stasis and treating muscle strain; its fruit, leaves, and root contain aromatic oil and safrole oleoresin; and it is also an ornamental landscape garden tree because of its peculiar shape and colorful leaves and the beautiful tree form [7,8,10,11]. In 2017, *S. tzumu* was listed in the Reference List of Major Cultivated Precious Tree Species in China by State Forestry Administration [12]. Meanwhile, some previous studies reported that *S. tzumu* is at risk due to their declining population size that resulted from overexploitation and other threats [7,8]. Therefore, it is crucial to conserve this precious species via enlarging its genetic resources and unraveling its genetic backgrounds of genetic diversity and population structure.

In recent years, with the rapid advancement of sequencing technology, developing molecular markers for the genetic survey of our interested species has become much easier and more effective [13]. Simple sequence repeat (SSR), also known as microsatellite, is a short repetitive DNA sequence consisting of a core motif (typically 2–6 bp) and a flanking sequence together [14]. SSR can be commonly spotted in both coding and noncoding regions of eukaryotic genomes [14], and it is widely used in the studies of population genetics, breeding programs, and phylogeography on account of its codominance, the abundance of genotypic polymorphism, high versatility, and variation detection rate [15,16,17,18,19]. Previous studies of *S. tzumu* primarily focused on germplasm investigation, taxonomy, cultivation, and population ecology [8,9,20,21,22,23,24,25], whereas a few studies have assessed the genetic diversity and population structure of this species [7,26,27,28]. For details, Ding et al. [26] firstly clarified the low-level genetic diversity (*H*_E_ = 0.280) of three natural populations of *S. tzumu* in Hubei Province [29] using an isozyme marker; Jiang et al. [27] found a high-level genetic diversity (*H*_E_ = 0.84) and a moderate genetic differentiation among five wild populations in Tianmu Mountain with five different altitudes via employing 13 polymorphic SSRs developed from *S. randaiense* [7] and *Cinnamomum camphora* [30]; Liu et al. [28] recently identified 11,862 SNP loci in 13 relatively agminated populations of *S. tzumu* and observed a relatively low-level genetic diversity (*H*_E_ = 0.3004) and a relatively high degree of population differentiation (*F*_ST_ = 0.118). The level of genetic diversity, status of population differentiation, and population structure on a relatively large scale were still controversial in *S. tzumu* in these investigations. Obviously, there is still a lack of some alternative molecular markers such as the nSSR marker directly mined from the genomic sequencing data of *S. tzumu* to augment its genetic resources and ascertain its genetic backgrounds.

The procedures of SSR development for a target species in a traditional way mainly consist of three steps: SSR discovery, primer design, and polymorphic survey in representative population or individuals. It is costly and extremely labor-intensive, time-consuming in the traditional experimental screening of the SSR polymorphic status and its subsequent applicability to genetic studies. Compared with the conventional methods or tools, CandiSSR is an easy-to-use, economic, and labor- and time-saving pipeline, which can friendly enable users to find putative polymorphic SSRs not only from the transcriptome datasets but also from multiple assembled genome sequences of a target species or genus along with several comprehensive assessments to help researchers focus more on subsequent genetic studies [19]. Hence, following Zhang et al.’s effective method of developing novel SSR markers for the interested species [31], we firstly selected six individuals from six geographically distant populations of *S. tzumu* for low-coverage whole-genome short-gun sequencing in our present study to directly develop polymorphic SSR markers. We further investigated the genetic diversity and population structure of these six wild populations of *S. tzumu* via the polymorphic nSSR loci screened and developed from the nuclear genome data of *S. tzumu*. The results here will provide useful genetic resources for further studies of population genetics, molecular breeding, and germplasm management of *S. tzumu* and its congeners, as well as lay a solid basis for proposing scientific conservation strategies and schemes for their better preservation.

## 2. Results

### 2.1. Prediction, Verification, and Transferability of Newly Developed nSSR Loci of S. tzumu

A total of 16,215 potential polymorphic nSSR loci with 424 repeated types were identified by the effective pipeline CandiSSR. The most abundant motif was dinucleotide repeat (12,248; 75.53%) followed by trinucleotide repeat (3203; 19.75%), and the tetranucleotide, pentanucleotide, and hexanucleotide repeats accounted for 2.61%, 0.91%, and 1.19%, respectively (Figure 1). According to Zhang et al.’s principles of primer screening [31], we selected 48 candidate polymorphic nSSRs to test their amplification success and effectiveness using six individuals of *S. tzumu* and then detected their polymorphisms via 12 individuals from six representative geographically distant populations (two individuals were randomly chosen from each population). As a result, 35 nSSR loci (73.92%) were successful in amplification, among which 27 pairs of primers (56.25%) were polymorphic. The sequences of these above 27 polymorphic nSSRs were submitted to GenBank and annotated against the GenBank nonredundant database using BLASTX [32]. The detailed characteristics of these nSSRs are shown in Table 1 and Appendix A. 

For the assessment of cross-species transferability of these 27 newly developed nSSR loci in other two *Sassafras* species (*S. albidum* and *S. randaiense*), we found that 23 nSSR loci (85.19%) were successfully amplified in *S. albidum* and *S. randaiense*. Two nSSRs (N362 and N2716) failed to yield positive amplicon, and the other two nSSRs (N2689 and N7047) did not show clear readable allelic bands in these two species. Furthermore, 17 nSSRs (62.96%) presented the polymorphisms in 22 individuals of *S. albidum*, and 14 nSSRs (51.85%) successfully presented the polymorphisms (various genotypes in different individuals) in 10 individuals of *S. randaiense* (Appendix A).

### 2.2. Genetic Diversity of nSSRs of S. tzumu

As a result, a total of 125 reproducible alleles were screened in 27 nSSR primers set by 136 individuals of *S. tzumu* from six populations (Appendix A). For each locus, the average of the number of alleles (*N*_A_), effective number of alleles (*N*_E_), observed heterozygosity (*H*_O_), and expected heterozygosity (*H*_E_) was 4.630, 1.750, 0.333, and 0.408, respectively. The polymorphism information content (*PIC*) of these loci varied from 0.071 (N1037) to 0.567 (N15402), and the average of *PIC*, *H*_S_, *H*_T_, and *I* was 0.356, 0.325, 0.413, and 0.742, respectively (Table 2). The proportion of nSSRs with moderate (0.25 < *PIC* < 0.5) or high polymorphism (*PIC* > 0.5) was 88.89%. These findings indicated that the majority of our developed nSSRs showed relatively high polymorphism in *S. tzumu*. The average inbreeding coefficient (*F*_IS_) was 0.209, and the *F*_IS_ in 21 loci were all greater than zero (Table 2).

For analyzing the genetic diversity of six *S. tzumu* populations, 11 loci (N43, N168, N711, N1037, N2689, N2799, N6336, N6882, N8706, N13148, and N16055) were excluded from subsequent calculation because of the excessive null allele frequency (greater than 0.08) [33] or significant linkage disequilibrium (less than 0.05) [34] according to the estimated results we obtained (Appendix A). The averages of *H*_O_, *H*_E_, *H*_S_, *H*_T_, *PIC*, *I*, and *F*_IS_ were changed to 0.359, 0.430, 0.324, 0.436, 0.370, 0.764, and 0.166 from 0.333, 0.408, 0.325, 0.413, 0.356, 0.742, and 0.209, respectively (Table 2). Overall, no significant changes were shown.

At the population level, the expected heterozygosity (*H*_E_) ranged from 0.195 (LL) to 0.387 (BYS), and the observed heterozygosity (*H*_O_) ranged from 0.244 (LL) to 0.464 (JGS). The highest values of allele richness (*A*_R_), number of effective allele (*A*_E_), average number of allele (*A*), number of private allele (*A*_P_), Shannon’s information index (*I*), and expected heterozygosity (*H*_E_) were all detected in the BYS population (2.891, 1.800, 3.125, 7, 0.674, and 0.387), and the lowest were in the LL population (2.016, 1.382, 2.125, 1, 0.341, and 0.195) (Table 3). Moreover, a genetic bottleneck effect was observed in population YX and JGS (Appendix A).

### 2.3. Population Structure and Differentiation in S. tzumu

High genetic differentiation (*F*_ST_ = 0.286) existed among six populations in *S. tzumu* [35]. The results of the AMOVA revealed that the main variation of *S. tzumu* was within the populations (71.60%), and 28.40% of the total variation was attributed to differences among the six populations (Table 4). The pairwise population genetic differentiation index (*F*_ST_) and gene flow (*N*_m_) among populations were also estimated in Table 5, from which we can find that all the *F*_ST_ values were significantly greater than 0.25 between population LL and the other five populations with the range of 0.356 (BYS) to 0.536 (JGS); and the values of *N*_m_ were all less than 1 among LL and the other five populations. The results of the Mantel test of the IBD effect revealed a significant correlation (*r* = 0.742, *p* = 0.006) between the geographical and genetic distance among the six populations (Appendix A).

STRUCTURE yielded the highest likelihood when individuals were grouped into two clusters (*K* = 2) with the maximum Δ*K* (Figure 2). However, the maximum of ln*p* (D) was exhibited with *K* = 5 (Figure 2), and the further strong population subdivision appeared from *K* = 2 to *K* = 5 (Figure 2). LL, the westernmost population, was assigned to one cluster (the red color in Figure 2), and the other five populations were assigned to another cluster (the green color in Figure 2) when running STRUCTURE with *K* = 2, although the population BYS appeared mixed into two clusters. When *K* = 3, populations BYS and HNS (the southernmost population) were formed as a separate cluster (the blue color in Figure 2). With *K* = 4, populations JZF and JGS (the northernmost population) were assigned to a separate cluster (the yellow color in Figure 2), whereas population YX (the easternmost population) was separated and showed some degree of mixed ancestry (the green color in Figure 2). When *K* = 5, the population BYS separated into another cluster (the purple color in Figure 2). 

Consistent with the result of STRUCTURE when *K* = 2, the topology of UPGMA and NJ trees all indicated that the six populations of *S. tzumu* could be grouped into two clades: population LL and the other remaining five populations (BYS, HNS, YX, JZF, and JGS) (Figure 3). Moreover, the first two components of the principal coordinate analysis (PCoA) were generated based on the mean genotypic genetic distance among the six populations of *S. tzumu*, accounting for 73.31% (Axis 1) and 11.58% (Axis 2) of the total variance. It presented that the JZF and JGS populations had the most similar value, and the other four populations formed their own positions (Figure 4). The above results also accorded with the results of phylogenic and genetic cluster analysis.

## 3. Discussion

### 3.1. Characterization, Polymorphism, and Transferability of Newly Developed nSSRs of S. tzumu

In this study, following Zhang et al.’s effective method and under the help of the pipeline of CandiSSR [31], we successfully predicted plenty of putative polymorphic nSSR loci from the sequenced nuclear genome’s contigs and scaffolds of six geographically distant individuals of *S. tzumu*. Compared with the traditional way of developing SSR markers, the above procedures are believed to be much more time- and energy-saving.

For the details of our candidate polymorphic nSSR loci for *S. tzumu*, we found that dinucleotide repeats were the most abundant type (75.53%) followed by the trinucleotide repeats (19.75%) (Figure 1). This distribution was commonly found in other tree species, such as *Dalbergia hupeana* (dinucleotide repeats account for 69.67%, whereas trinucleotide repeats account for 27.03%); and *Parrotia subaequalis* (80.54%, 16.77%) [36,37]. Worth mentioning is the fact that the trinucleotide repeat is deemed to be one of the most abundant repetition motif types for tree or fruit species because they hold less chance to the frameshifts in the coding sequences of species [36,38]. Meanwhile, a relatively low SSR density was observed in the coding sequences of species because the high mutation rate of SSR may affect the gene function, regulation of gene transcription, translation, and metabolic activities [38]. Indeed, the obtained 27 polymorphic nSSRs were annotated against the GenBank nonredundant database, among which, seven nSSR loci were correspondingly linked to transcription and other vital activities (Table 1). In addition, for the assessment of the polymorphism of these 27 novel nSSRs, the majority of the developed nSSRs (88.89%) showed high or moderate polymorphism (*PIC* > 0.25), which indicated that these nSSRs have great power to illuminate the population genetic diversity and variation of *S. tzumu*.

Furthermore, many previous studies indicated that SSR markers have high cross-species transferability among closely related species in plants [7,36,37,39]. For the genus *Sassafras* here, 20 nSSRs from *S. randaiense* has been successfully transferrable to *S. tzumu* using one individual [7]; in our study, the tests of interspecies transferability of the 27 nSSRs from *S. tzumu* to the other two *Sassafras* species yielded a transfer success rate of 85.19% (23 loci), and the proportion of polymorphic primers for *S. albidum* and *S. randaiense* was relatively high (62.96% and 51.85%, respectively) (Appendix A). It revealed a relatively high cross-species transferability of these nSSR markers, which will have great assistance in the studies of population genetics, phylogeography, phylogenetics, molecular breeding, and resource management of *S. tzumu* and the other Sassafras (Lauraceae) species.

### 3.2. Population Genetic Diversity of S. tzumu

Extensive literature pointed out that plant genetic diversity could be affected by various internal and external factors; for internal factors, they include the breeding system, genetic drift (bottleneck effect as an exceptional case), natural selection, gene mutation, and gene flow; whereas the environmental change and human interference are supposed to be the external factors [40,41,42,43,44,45,46,47,48,49,50]. The expected heterozygosity (*H*_E_) and Shannon’s information index (*I*) are commonly used to assess population genetic diversity [36]. In this study, *H*_E_ and *I* were estimated to be 0.317 and 0.527, respectively, at the population level (Table 3), which were higher than the three populations in Hubei Province in China of *S. tzumu* (*H*_E_ = 0.280) uncovered by isozyme analysis [26], but significantly lower than the five populations in Tianmu Mountain along with different altitudes (*H*_E_ = 0.802) revealed by SSR markers [27]. We assumed that these differences might be due to different methods and sample distributions [51]. At the species level, our results revealed an intermediate level of genetic diversity (*H*_E_ = 0.430, *I* = 0.764) (Table 2), which was consistent with the study of Liu (*H*_E_ = 0.3004) [28]. Meanwhile, this genetic diversity calculated by our study with more extensive sampling covering the distribution range of *S. tzumu* was lower than the average *H*_E_ of long-lived perennial plants (*H*_E_ = 0.68), but slightly higher than the average of endemic plants (*H*_E_ = 0.42) [41]. Additionally, compared with the other species of Lauraceae these share the similar distribution range of *S. tzumu*, the genetic diversity of *S. tzumu* in this study was generally lower than *Machilus pauhoi* (*H*_E_ = 0.751) [52], *Lindera glauca* (*H*_E_ = 0.59, *I* = 1.17) [53], *Cinnamomum camphora* (*H*_E_ = 0.785) [54], and *Persea americana* (*H*_E_ = 0.6927) [55] but a little higher than those of some endangered species, such as *Litsea szemaois* (*I* = 0.3826 ± 0.2333) [56], *Cinnamomun chago* (*H*_E_ = 0.340) [57], and *Phoebe bournei* (*H*_E_ = 0.43, *I* = 0.72).

Considering that *S. tzumu* is a species with entomophily, and the main factor affecting genetic diversity is the breeding system [27,35,40,44,58,59], therefore, *S. tzumu* has a lower gene flow than plants with anemophily [40]. Furthermore, the characteristics of pollen abortion, short time to maintain pollen viability, a low proportion of normal-growing embryo sacs, and low seed setting percentage of *S. tzumu* [9] also lead to a lower level of gene flow [40] (*N*_m_ = 0.625) (Table 5). Moreover, the relatively high level of inbreeding (*F*_IS_ = 0.166) and heterozygote deficit were detected in *S. tzumu*, which also resulted in its low genetic diversity. Furthermore, two populations (YX and JGS) were examined experiencing a recent genetic bottleneck effect [60]; we assumed that human interference and habitat fragmentation were the significant affect factors for this result according to the field investigation. As indicated by a previous study, the total distribution areas of *S. tzumu* have migrated southward and decreased by 15.06% from the Last Glacial Maximum (LGM) to present [61]. LL, as the westernmost population at the edge of the distribution areas, was located in the distribution area with contraction from the LGM to present, which corresponded to the result of the lowest genetic diversity of LL among the six populations (Table 3). Some species in partially overlapping distribution areas with *S. tzumu* also had low genetic diversity due to the above factors, such as *Salix taishanensis* (*I* = 0.3505) [62], *Magnolia sinostellata* (*H*_E_ = 0.27312) [63], *Michelia chapensis* (*H*_E_ = 0.3255) [64], and *Sinomanglietia glauca* (*H*_E_ = 0.423) [65].

### 3.3. Population Structure and Differentiation of S. tzumu

The population genetic structure has an inseparable relationship with the biological evolution process, which included natural selection in heterogeneous environments, genetic drift, and gene flow [66,67]. Structure analysis in this study optimally divided six selected *S. tzumu* populations into two clusters: LL, the westernmost population in the Yunnan–Guizhou Plateau, formed a separate cluster, whereas the other five populations (BYS, HNS, YX, JZF, and JGS) were assigned to another cluster (Figure 2). Moreover, there are many mountains in subtropical China [68] based on the study about the distribution of the mountain ranges and peaks in China [69], and the significant effect of isolation by distance (IBD) was detected by the Mantel test (*r* = 0.742, *p* = 0.006) in this study (Appendix A), which corresponded to the relatively large geographic distance and barriers of multiple mountain ranges of the six populations of *S. tzumu*, and these two factors also played facilitating roles in promoting genetic differentiation and hindering the gene flow. A previous study also revealed the effect of IBD and the barrier of the Meiling Mountain and Lu Mountain in *S. tzumu* [28]. In the present study, we identified that the Wuling Mountain, Dalou Mountain, and Yunnan–Guizhou Plateau played major barrier roles among LL and the other five populations. Similar mechanism of population structures has been reported in other species, such as *Cerasus dielsiana* (Wuling Mountain) [70]; *Tetracentron sinense* (Qinling Mountain, Hengduan Mountain, and Dalou Mountain) [35]; *Sinomanglietia glauca* (Wuling Mountain) [65]; and *Phoebe zhennan* (Wu Mountain and Wuling Mountain) [71].

When it comes to the level of population differentiation, our study unveiled a relatively high genetic differentiation (*F*_ST_ = 0.286) that existed among the six populations of *S. tzumu*. Compared with other forest trees of Lauraceae, it was much higher than *Cinnamomun camphora* (*F*_ST_ = 0.109) [54], *Cinnamomun chago* (*F*_ST_ = 0.2198) [57], and *Cryptocarya chinensis* (*F*_ST_ = 0.141) [72] but lower than *Lindera glauca* (*F*_ST_ = 0.362) [73] and *Machilus pauhoi* (*F*_ST_ = 0.3366) [74], and had a similar value with *Phoebe bournei* (*F*_ST_ = 0.287) [75]. In addition, the AMOVA indicated that 28.40% (Table 4) of the total variation was distributed among the populations, and most of the genetic variation was distributed within the populations. This finding agreed that long-lived, outcrossing, and/or late successional taxa retain most of their genetic variability within populations with the analyses based on microsatellite DNA loci [41]. Apart from the geographic distance, Liu pointed out that the environmental distance had a more significant effect on the genetic distance than the geographic distance (IBE = 1.54 IBD), especially with the influence of the mean annual temperature [28], and the amount of precipitation was also considered to be the main effect factor on the suitability degree of the distribution areas of *S. tzumu* [61]. These findings supported our views and were consistent with the biological character of habitat preference with warmth and wetness of *S. tzumu* [28]. Therefore, both the geographic distance and climate difference played important roles for the intraspecific divergence of *S. tzumu*.

### 3.4. Conservation Implications for S. tzumu

Above all, a moderately low level of genetic diversity is believed to relatively lack the ability to adapt to various habitats and changing climate [28,66], and increase the risk of being endangered for a species [7,8]. Here, in our study, *S. tzumu* presented an intermediate level of genetic diversity in its distribution range. Hence, it is urgent and crucial to protect this Arctic Tertiary relict endemic species in China. As revealed by our study herein, population LL (the westernmost population) and the other five populations (BYS, HNS, YX, JZF, and JGS) of *S. tzumu* formed two detached clusters, which could be considered as two ESUs for its long-term preservation of genetic resources. What is more, all populations should be protected in situ based on the low genetic diversity among populations as resources (money, manpower, etc.) permit.

## 4. Materials and Methods

### 4.1. Plant Materials and DNA Extraction

The fresh young leaves of six individuals were collected from six geographic representative populations in southern China (Figure 2, Table 6) to represent the various distributions of *S. tzumu*. To evaluate and validate the polymorphism of the nSSRs developed from our sequencing data, the leaf material of two individuals in *S. tzumu* was collected from each of the six populations from across the distribution range of the species in southern China. For each population, more than 14 individuals were sampled at least 10 m apart to minimize the likelihood of sampling clones, resulting in a total of 136 individuals (Table 6). The polymorphic nSSRs were further tested for transferability to two related species in *Sassafras* genus, *S. albidum* and *S. randaiense* (Appendix A). The leaf materials of 22 individuals of *S. albidum* were collected in 22 different regions in the USA and 10 individuals of *S. randaiense* in Taiwan, China (Appendix A). The collected leaf materials were dried and stored in silica gel at 4 °C for further DNA extraction. The representative voucher specimens of the *Sassafras* species were deposited in the Herbarium of Nanjing University Table 6 and Appendix A). Total genomic DNA was extracted from leaf tissue using Plant DNAzol Reagent (Invitrogen Life Technologies), following the manufacturer’s protocol. DNA quality was examined on a 0.8% agarose gel stained with 1× GelRed (Biotium Inc., Fremont, CA, USA), and the concentration was checked using a Nano-Drop Lite spectrophotometer (Thermo Fisher Scientific, Waltham, MA, USA).

### 4.2. NSSR Development, Polymorphism, and Transferability Assessment

One individual was selected from each of the 6 populations to construct six next-generation sequencing genomic paired-end DNA libraries using the low-coverage shotgun sequencing in the HiSeq XTen platform (Illumina Inc., San Diego, CA, USA) at BGI (China). For each individual, about 7 Gb of raw data was obtained with paired-end 150 bp read length. Following Zhang et al.’s effective method [31], we filtered the chloroplast- and mitochondria-containing contigs and used CandiSSR [19] to identify potentially polymorphic nSSR primer pairs from the remaining scaffolds of the assembled nuclear genome database with a individual (from the BYS population) as the reference. Then, the specific potential polymorphic nSSRs were selected based on the above method [31]. The total selected candidate nSSRs were used for preliminary screening, and the corresponding primers were synthesized by Sangon Biotech (Shanghai, China).

Six randomly selected individuals from six populations were used to initially screen the amplification success and effectiveness of the primer pairs and verify the optimal primer temperature by PCR amplifications and 1% agarose gel stained with 1× GelRed (Biotium Inc., Fremont, CA, USA). The polymorphism of each locus was then performed by selecting 12 individuals from 6 representative geographically distant populations (two individuals were chosen from each population at random) (Table 6). Cross-species amplification was then conducted in two more *Sassafras* species (Appendix A) to test the transferability of the polymorphic nSSRs above. PCR amplifications were performed on a GeneAmp9700 DNA Thermal Cycler (Perkin-Elmer, Waltham, MA, USA) following the standard protocol of the AmpliTaq Gold 360 Master PCR kit (Thermofisher Biotech Company, Applied Biosystems, Foster City, CA, USA) in a final volume of 15 µL, which contained 1 µL of template DNA, 7.5 µL of AmpliTaq Gold 360 Master Mix (Thermofisher Biotech Company, Applied Biosystems, Foster City, CA, USA), 5.5 µL of deionized water, 0.5 µL of reverse primers (10µM), and 0.5 µL of forward primers (10µM), with the 5′ end labeled with a fluorescent dye (FAM, HEX, TAMRA, or ROX) (Appendix A). Moreover, the PCR procedure was as follows: 5 min initial denaturation at 95 °C, 35 cycles of 45 s at 95 °C, 30 s annealing at an optimal primer temperature (Appendix A) and 30 s synthesis at 72 °C, followed by a final 10 min extension step at 72 °C and a 4 °C maintained temperature. The PCR products were detected by an ABI 3730XL capillary electrophoresis analyzer (Applied Biosystems, Foster City, CA, USA) with GeneScan-500LIZ as an internal reference (Applied Biosystems). Fragments were genotyped for their presence or absence at each locus, and the allele sizes were scored by using GeneMarker v2.2.0 (SoftGenetics, State College, PA, USA) and manually checked twice to reduce genotyping errors.

### 4.3. Assessment of Genetic Diversity of S. tzumu

For each nSSR locus, the polymorphism information content (*PIC*), number of alleles (*N*_A_), expected heterozygosity (*H*_E_), and observed heterozygosity (*H*_O_) were calculated by CERVUS v3.0.7 [76]; the total genetic diversity (*H*_T_) and average genetic diversity within populations (*H*_S_) were estimated by FSTAT v2.9.4 [77]; the inbreeding coefficient (*F*_IS_) was calculated by INEst v2.2 [78]; and the Shannon’s information index (*I*) and effective number of alleles (*N*_E_) were counted by Popgene v1.31 [79]. Additionally, deviation from the Hardy–Weinberg equilibrium (HWE) and linkage disequilibrium (LD) [80] for each locus was tested using a Markov chain (dememorization: 10,000; batches: 20, iterations per batch: 5000) in GENEPOP v4.7.5 [81]. The null allele frequency of each locus in each population was estimated based on the expectation maximization algorithm in FREENA [82].

For each population of *S. tzumu*, the allelic richness (*A*_R_) was estimated by FSTAT v2.9.4 [76], and the following genetic diversity parameters were calculated by GenALEx v6.502 (Canberra, Australia) [83]: the average number of alleles (*A*), effective number of alleles (*A*_E_), number of private alleles (*A*_P_), observed heterozygosity (*H*_O_), expected heterozygosity (*H*_E_), unbiased expected heterozygosity (*μH*_E_), and Shannon’s information index (*I*). Furthermore, the Wilcoxon signed-rank test and Z-test were used to estimate the excess of heterozygosity under the infinite model (IAM), strict stepwise mutation model (SMM), and two-phase model (TPM) by INEst v2.2 [77] to perform a bottleneck test [84] for each population.

### 4.4. Population Structure and Differentiation of S. tzumu

The total genetic differentiation coefficient (*F*_ST_) and pairwise population genetic differentiation index (*F*_ST_) were calculated by Genetix v4.05 program [85], and an estimate of *N*_m_ (the number of migrants per generation/gene flow) among the populations was computed using the formula *N*_m_ = 0.25(1 − *F*_ST_)/*F*_ST_ [86]. In addition, the analysis of molecular variance (AMOVA) was performed to partition the total genetic variance among populations using ARLEQUIN v3.5.2.2 [87]. Moreover, GenALEx v6.502 (Canberra, Australia) [83] was used to compute the principal coordinate analysis (PCoA) and perform the Mantel test to estimate the correlation between Nei’s unbiased genetic distance and geographic distance (9999 permutations) to reveal the influence of the isolation-by-distance (IBD) model. Moreover, the study of the phylogenetic relationship among the populations of *S. tzumu* was performed by POPTREE [88] based on shared allele distances among the populations (*D*_A_ genetic distance) to build the UPGMA and NJ tree [89] using unweighted pair-group method (UPGMA) [90] and neighbor-joining method (NJ) [91].

Bayesian methods were utilized to reveal the optimal number of genetic clusters (*K*) for the studied populations, and these Bayesian analyses were implemented using STRUCTURE v2.3.4 [92], with ten independent runs for each *K* value that ranges from 1 to 10 that were performed to determine the maximum value of the posterior probability of the data [ln*p*(D)] with 100,000 MCMC (Markov Chain Monte Carlo) repetitions and 100,000 burn-in periods. We assumed that the studied populations conformed to the admixture model and independent allele frequencies. The obtained results from STRUCTURE were further interpreted by STRUCTURE HARVESTER [80], which implemented Evanno’s Δ*K* method [93] to estimate the parameter (Δ*K*) that corresponded to the change of ln*p*(D) between consecutive *K* values for the calculation of the optimal *K*. The most likely number of cluster (*K*) was identified as the one that maximized ln*p*(D) and/or Δ*K*.

## 5. Conclusions

In this study, we first identified 16,215 potential polymorphism nSSR loci based on the assembled nuclear genome database of six individuals of *S. tzumu*. The di- (75.53%) and trinucleotide (19.75%) repeats were the most abundant. A total of 27 novel nSSRs with polymorphism were utilized to illuminate the genetic diversity of 136 individuals in six populations of *S. tzumu*, resulting most markers (88.89%) showing high or moderate polymorphism (*PIC* = 0.356). The majority of nSSRs (85.19%) could be successfully transferred to the other two *Sassafras* species, and some of them showed polymorphism. The population genetic results revealed a moderately low genetic diversity of *S. tzumu* with *H*_E_ ranging from 0.195 (LL) to 0.387 (BYS) at the population level, and the average *H*_E_ was 0.430 at the species level. The bottleneck effect appeared in some populations (YX and JGS). The population differentiation was high (*F*_ST_ = 0.286), and the majority of variations (71.60%) were within the populations. Consistent with the results of the principal coordinate analysis (PCoA) and phylogenetic trees, structure analysis optimally divided these six *S. tzumu* populations into two clusters, and the further strong population subdivision appeared from *K* = 2 to *K* = 5, which corresponded to two evolutionarily significant units (ESUs). Moreover, the significant correlation between genetic and geographic distance was tested by the Mantel test (*r* = 0.742, *p* = 0.006), clarifying the effect about isolation by distance (IBD), which could be possibly explained by the low gene flow (*N*_m_ = 0.625), a relatively high degree of inbreeding (*F*_IS_ = 0.166), a relatively large distribution, and mountainous barriers. Taken together, our research gives a deeper understanding of the population genetic diversity and structure of *S. tzumu* in subtropical China and will contribute to the conservation and utilization of *S. tzumu* and the other *Sassafras* (Lauraceae) species.

## Figures and Tables

**Figure 1 plants-11-02706-f001:**
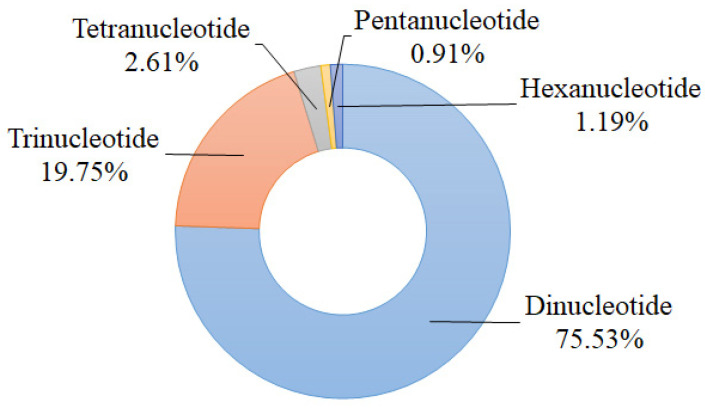
Statistics of types of candidate polymorphic nSSRs of *S. tzumu*.

**Figure 2 plants-11-02706-f002:**
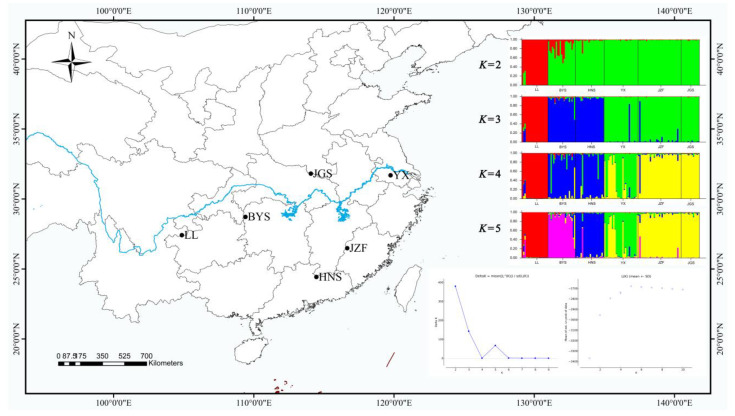
Geographic distribution of six populations of *S. tzumu* in China and population genetic structure among six populations of *S. tzumu* estimated using STRUCTURE based on nSSR with values of *K* from 2 to 5.

**Figure 3 plants-11-02706-f003:**
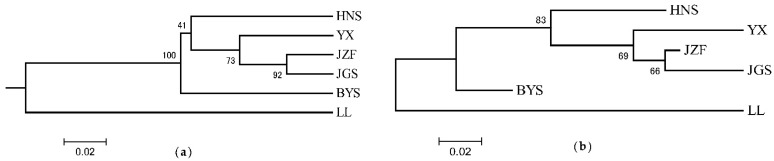
Phylogeny relationships among six populations of *S. tzumu* by unweighted pair-group method (UPGMA) (**a**) and neighbor-joining (NJ) method (**b**) based on *D*_A_ distance.

**Figure 4 plants-11-02706-f004:**
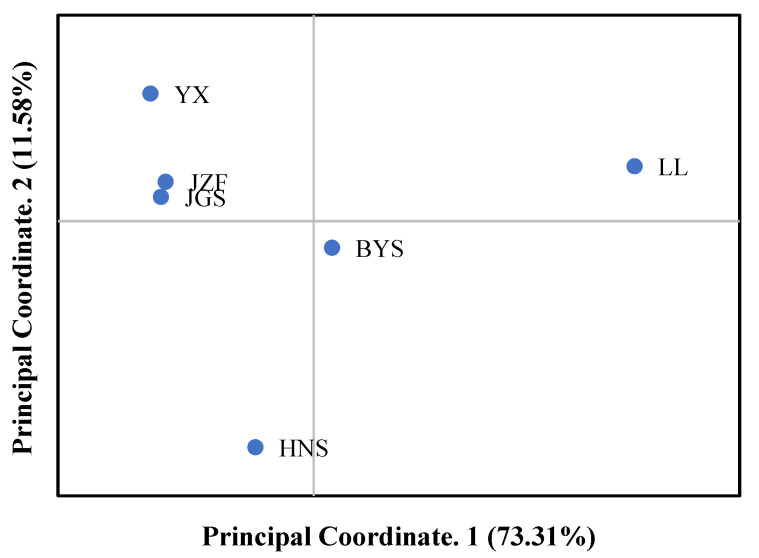
Principal coordinate analysis (PCoA) among six populations based on mean genotypic genetic distance.

**Table 1 plants-11-02706-t001:** Characteristics of 27 newly developed nSSR loci of *S. tzumu* in this study.

Locus	Repeat Motif	Optimal Primer Temperature (°C)	Size Range (bp)	GenBank Accession Number	BLASTX Top Hit Description
N43	(AAAG)_8_	59	157–169	OP094707	zinc finger protein [Cinnamomum micranthum f. kanehirae]
N168	(AAC)_7_	56	134–146	OP094689	-
N362	(AAG)_7_	55	185–206	OP094695	-
N711	(AATAAA)_5_	59	187–223	OP094710	putative nuclease HARBI1 [Cinnamomum micranthum f. kanehirae]
N1037	(ACAA)_5_	55	139–155	OP094684	hypothetical protein SLEP1_g25016 [Shorea leprosula]
N1047	(ACAT)_5_	56	162–166	OP094703	-
N2689	(AGA)_8_	52	188–197	OP094687	Alcohol dehydrogenase superfamily, zinc-type [Cinnamomum micranthum f. kanehirae]
N2716	(AGAT)_5_	59	159–175	OP094697	-
N2799	(AGGTGA)_5_	52	159–183	OP094690	-
N6336	(ATC)_6_	59	178–187	OP094701	-
N6339	(ATC)_6_	56	169–172	OP094694	-
N6493	(ATGT)_6_	59	128–148	OP094692	-
N6882	(CAC)_8_	55	167–188	OP094708	-
N6889	(CACCTC)_5_	56	159–189	OP094705	-
N7047	(CCCTCA)_5_	59	182–206	OP094702	-
N8706	(CTC)_6_	55	114–129	OP094688	-
N8723	(CTC)_9_	51	156–177	OP094685	zinc finger protein 1-like protein [Cinnamomum micranthum f. kanehirae]
N8865	(CTT)_8_	55	191–203	OP094699	methyltransferase-like protein 13 isoform X1 [Cinnamomum micranthum f. kanehirae]
N10640	(GAT)_6_	53	181–187	OP094691	N-carbamoylputrescine amidase [Glycine max]
N13148	(TATAC)_5_	56	162–177	OP094704	-
N13155	(TATG)_5_	58	179–191	OP094686	-
N13174	(TATTTT)_6_	56	178–196	OP094706	-
N14929	(TCT)_7_	57	187–202	OP094698	-
N15402	(TGA)_6_	57	178–199	OP094709	-
N15940	(TTC)_5_	55	82–100	OP094700	-
N16055	(TTCT)_5_	55	161–169	OP094693	-
N16204	(TTTTC)_5_	56	133–148	OP094696	-

**Table 2 plants-11-02706-t002:** Genetic characteristics of 27 polymorphic nSSRs in *S. tzumu*.

Locus	*N*_A_ ^1^	*N*_E_ ^2^	*H*_O_ ^3^	*H*_E_ ^4^	*H*_S_ ^5^	*H*_T_ ^6^	*PIC* ^7^	*I* ^8^	*F*_IS_ ^9^
N43	4	1.858	0.537	0.464	0.446	0.464	0.384	0.759	−0.159
N168	4	1.950	0.412	0.489	0.406	0.487	0.402	0.792	0.158
N362	7	1.758	0.228	0.433	0.222	0.453	0.400	0.878	0.474
N711	7	1.447	0.250	0.310	0.288	0.318	0.291	0.676	0.194
N1037	5	1.077	0.051	0.072	0.064	0.067	0.071	0.201	0.285
N1047	2	1.826	0.676	0.454	0.420	0.452	0.350	0.645	−0.493
N2689	4	1.755	0.397	0.432	0.375	0.440	0.393	0.795	0.081
N2716	5	1.593	0.088	0.374	0.142	0.403	0.328	0.689	0.765
N2799	5	1.996	0.426	0.501	0.308	0.503	0.418	0.853	0.149
N6336	4	1.322	0.066	0.244	0.228	0.252	0.228	0.500	0.730
N6339	2	1.534	0.257	0.349	0.287	0.358	0.287	0.532	0.264
N6493	6	2.199	0.728	0.547	0.526	0.550	0.448	0.929	−0.332
N6882	6	1.797	0.265	0.445	0.381	0.440	0.426	0.985	0.406
N6889	5	2.189	0.551	0.545	0.438	0.538	0.440	0.891	−0.012
N7047	4	1.494	0.287	0.332	0.297	0.329	0.303	0.635	0.136
N8706	5	2.273	0.507	0.562	0.519	0.561	0.511	1.067	0.098
N8723	7	1.655	0.279	0.397	0.275	0.417	0.379	0.893	0.297
N8865	4	1.676	0.419	0.405	0.327	0.393	0.359	0.731	−0.035
N10640	3	2.301	0.199	0.567	0.223	0.568	0.469	0.905	0.651
N13148	5	1.547	0.331	0.355	0.340	0.372	0.327	0.692	0.068
N13155	3	1.956	0.449	0.491	0.349	0.493	0.373	0.700	0.086
N13174	4	1.533	0.265	0.349	0.316	0.367	0.320	0.682	0.242
N14929	4	1.859	0.434	0.464	0.358	0.436	0.365	0.703	0.065
N15402	7	2.532	0.353	0.607	0.470	0.613	0.567	1.246	0.420
N15940	6	1.391	0.294	0.282	0.255	0.310	0.266	0.611	−0.043
N16055	3	1.355	0.007	0.263	0.241	0.283	0.237	0.487	0.972
N16204	4	1.391	0.235	0.282	0.274	0.296	0.258	0.554	0.166
T-Mean ^10^	4.630	1.750	0.333	0.408	0.325	0.413	0.356	0.742	0.209
E-Mean ^11^	4.563	1.805	0.359	0.430	0.324	0.436	0.370	0.764	0.166

^1^ Number of alleles; ^2^ Effective number of alleles; ^3^ Observed heterozygosity; ^4^ Expected heterozygosity; ^5^ Average genetic diversity within populations; ^6^ Total genetic diversity; ^7^ Polymorphism information content; ^8^ Shannon’s information index; ^9^ inbreeding coefficients; ^10^ Mean of total loci; ^11^ Mean of loci after excluding the loci with excessive null allele frequencies or significant linkage disequilibrium.

**Table 3 plants-11-02706-t003:** Genetic parameters of six populations in *S. tzumu*.

Population Code	*N* ^1^	*A* ^2^	*A*_E_ ^3^	*A*_R_ ^4^	*A*_P_ ^5^	*H*_O_ ^6^	*H*_E_ ^7^	*μH*_E_ ^8^	*I* ^9^
BYS	21	3.125	1.800	2.891	7	0.366	0.387	0.396	0.674
LL	20	2.125	1.382	2.016	1	0.244	0.195	0.200	0.341
HNS	22	2.750	1.643	2.575	5	0.318	0.355	0.363	0.599
YX	26	2.438	1.648	2.258	1	0.392	0.322	0.329	0.526
JZF	33	2.563	1.555	2.213	3	0.381	0.309	0.313	0.502
JGS	14	2.125	1.595	2.125	2	0.464	0.334	0.346	0.518
Mean/Total	136	2.521	1.604	2.346	19	0.361	0.317	0.325	0.527

^1^ Number of individuals; ^2^ Average number of alleles; ^3^ Effective number of alleles; ^4^ Allelic richness (average of per locus in each population) based on minimum sample size of population with 14 diploid individuals; ^5^ Number of private alleles; ^6^ Observed heterozygosity; ^7^ Expected heterozygosity; ^8^ Unbiased expected heterozygosity; ^9^ Shannon’s information index.

**Table 4 plants-11-02706-t004:** Result of AMOVA for *S. tzumu*.

Source of Variation	d.f.	Sum of Squares	Variance Components	Percentage of Variation
Among populations	5	242.774	1.027 Va	28.40
Within populations	266	689.178	2.591 Vb	71.60
Total	271	931.952	3.618	

**Table 5 plants-11-02706-t005:** Pairwise population genetic differentiation index (*F*_ST_) (above diagonal) with all significant population differentiation where *p* < 0.01 and number of migrants per generation (*N*_m_) (below diagonal) among populations of *S. tzumu*.

	BYS	LL	HNS	YX	JZF	JGS
BYS	-	0.356	0.147	0.234	0.184	0.205
LL	0.452	-	0.443	0.530	0.514	0.536
HNS	1.448	0.314	-	0.224	0.185	0.182
YX	0.820	0.221	0.868	-	0.127	0.162
JZF	1.109	0.236	1.102	1.712	-	0.063
JGS	0.970	0.216	1.120	1.292	3.740	-
Total *F*_ST_	0.286					
Total *N*_m_	0.625					

**Table 6 plants-11-02706-t006:** Information of plant materials of *S. tzumu* collected in this study.

Population Code	Collection Locality	Geographic Coordinates	Number of Individuals	Voucher Specimens
BYS	Baiyun Mountain Natural Reserve, Hunan Province, China	28.7301° N, 109.3955° E	21	ZJB201903
LL	Leli Village Forestry Farm, Yunnan Province, China	27.4202° N, 104.8601° E	20	ZJB201904
HNS	Huangniushi Natural Reserve, Guangdong Province, China	24.4294° N, 114.4556° E	22	ZJB201905
YX	Yixing National Forest Park, Jiangsu Province, China	31.6986° N, 119.7449° E	26	ZJB201906
JZF	Junzifeng Peak Natural Reserve, Fujian Province, China	26.4863° N, 116.6559° E	33	ZJB201907
JGS	Jigong Mountain Natural Reserve, Henan Province, China	31.8185° N, 114.0583° E	14	ZJB201908

## Data Availability

The data that support the findings of this study are openly available at the National Center for Biotechnology Information (https://www.ncbi.nlm.nih.gov/genbank/) (access on 28 July 2022); The GenBank accession number is shown in Table 1.

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
