# Peer review of "Genetic Diversity and Population Structure of an Arctic Tertiary Relict Tree Endemic to China (Sassafras tzumu) Revealed by Novel Nuclear Microsatellite (nSSR) Markers"

_plants, 2022, doi:10.3390/plants11202706_

Round 1

Reviewer 1 Report

1. In this paper, it may have been more important to select polymorphic markers in S. tzumu and transferbility markers to S. albidum and S. randaiense. More explanations are needed as to why the transferbility marker was first selected.

2. The markers are written as Nuclear Microsatellite Markers in the title and nSSR in the manuscript. The use of the same word is required.

Reviewer 2 Report

This study developed SSR markers using the nucleus database of Sassafras tsumu, and the markers apply for evaluate genetic diversity and structure of 6 natural populations. This study successfully isolated nuclear SSR markers for satisfying analysis of genetic diversity and structure of focal species. This study will provide information for conservation and utilization of the Sassafras species.

Major points

(1)  At L216, the authors mentioned that population BYS was found most closely related to the population LL. But this is not true. Because LL is not monophyletic with BYS, LL is equivalently distant from BYS and other populations. PCoA indicates also that LL is far from all other populations, and LL and BYS are not closely related. 

(2)  In L292, the authors mentioned populations YX and JGS have been experienced bottleneck based on the significant results of bottleneck. Then they suggest that population reduction through the southward migration from LGM to current could be the possible reason. However, considering that the bottleneck test detects only a recent bottleneck event (less than 4Ne generations ago [Cornuet and Luikart 1996], this interpretation should be reconsidered.

Minor points

L166   This study only utilized tri and more nucleotide as the makers although most repeats were di-nucleotide. Is there any specific reason for the maker selection?

L28-   “We further…”, the sentence is not simple to understand. Please consider rewriting.

L32   Can the abbreviation of population be used in the abstract? Is it better not to use?

L193-   “non-empty group” is not understandable.

L213   “K = 2” the K should be Italic.

L359   “mor” should be “more”.
